# Performance of Generator Translation and Rotation on Stroke Length Drive of the Two-Rod Mechanism in Renewable Energy Power Plant

**Hendra Hendra** [1,2,*] , **Dhimas Satria** [1,2] , **Hernadewita Hernadewita** [3] , **Yozerizal Yozerizal** [4] , **Frengki Hardian** [5] and **Ahmed M. Galal** [6,7]

1 Department of Mechanical Engineering, Faculty of Engineering, University of Sultan Ageng Tirtayasa Banten, Cilegon 42435, Indonesia
2 Design Manufacture and Material Mechanic Laboratory, COE Petrokimia University of Sultan Ageng Tirtayasa Banten, Cilegon 42435, Indonesia
3 Magister of Industrial Engineering, University of Mercu Buana, Jakarta 10650, Indonesia
4 Mechanical Engineering, New Red White Company, Cikupa 15710, Indonesia
5 Magister of Notary, University of Yarsi, Jakarta 10510, Indonesia
6 Department of Mechanical Engineering, College of Engineering in Wadi Al Dawasir, Prince Sattam bin Abdulaziz University, Wadi Al Dawasir 11991, Saudi Arabia
7 Production Engineering and Mechanical Design Department, Faculty of Engineering, Mansoura University, El Mansoura 35516, Egypt
* Correspondence: hendra@untirta.ac.id

**Abstract:** Generators are the main components in renewable energy power plants, especially in plants powered by ocean waves. The generator consists of two components of translational and rotational motion. Generators of translational and rotational motion can produce electric power from renewable energy sources such as water, wind, sea waves, biomass, and others. The voltage and electric power are the performance values of the translational and rotational generators which are affected by the type of magnet, the number of coil windings, the distance between the magnet and the coil winding and rotation, the geometry of the drive components, the type of drive, the length of the generator drive stroke, and so on. The types of translational and rotational generator drives can be found in the use of pneumatic motion mechanisms, two-rod motion, crankshaft motion, and others. A common problem in older power plants was that generator components were heavy, easy to break, less rigid, and had low rotation speed. Therefore, to overcome this problem, a generator with a two-rod mechanism is used in this research. In this paper, the generator drive step using a two-rod motion mechanism is used to run the generator. The length of the piston stroke is used to determine the performance of the generator, set at a length of 170–270 mm. The results show that the generator with two-rod motion mechanism rotating at 100–250 rpm can produce 30.9–55 volts at a frequency of 6.9–63.7 Hz with a maximum power of 0.377 w. By setting a piston stroke length of 170 mm, we obtained a rotation of 100–191 rpm and an electrical voltage of 30.9−35 volts. At a piston stroke length of 230 rpm, a rotation of 78–172 rpm is obtained with an electrical voltage of 47.7–55.5 volts. A piston stroke length of 270 mm produces a rotation of 172–256.5 rpm with a mains voltage of 39.9–55.5 volts. Testing the generators of translational and rotational motion using a two-rod motion mechanism in series and parallel with a stroke length of 270 mm produced a rotation from 179.2 to 242.3 rpm and an electric voltage from 57.4 to 79.5 volts and become constant at 35.6 volts by using a parallel mechanism. These results show that the generator translation and rotation motion can produce electric power by using renewable energy resources.

**Keywords:** two-rod mechanism; generator; translation; rotation; voltage

## 1. Introduction

Renewable energy power plants are power plants that are urgently needed at this time as a substitute for fossil energy power plants [1–3]. Renewable energy power plants are supported by natural resources that are always available and are environmentally friendly to the main sources of their energy, such as water [4,5], wind [6–11], solar [10,11], biomass [12,13], ocean and wave [14–33], etc. The high initial investment for renewable energy power plant components has forced researchers to search for alternative usages of components, processes, materials, etc., which are cheap and easy to use. One way is to utilize waste polyvinyl chloride (PVC) tubes [27–29] and other materials as part of a new renewable energy power plant component [31,32]. Energy sources for new renewable energy power plants can be obtained from solar energy, ocean energy, wind energy, ocean wave energy, biomass energy, and other sources of energy [4–33]. This energy can be converted into electrical energy from mechanical energy (kinetic energy and potential energy) by using several mechanical and electrical components, like generators and turbines.

Renewable energy power plant components consist of mechanical and electrical components including turbines, generators, batteries, generator housings, and turbines, as well as other components. The turbine is the component that drives the generator to produce electricity and is stored in the battery. In addition to turbines, generator drive components can use crankshaft mechanisms, two-rod mechanisms, pneumatic mechanisms, and other mechanisms.

The driving component of the generator is an important component for the generator's usefulness. The driving component serves to move the stator and rotor on the generator to produce an electric voltage. The movement mechanism of the driving components is part of the selection of the generator motion components, while the motion mechanism of the translational and rotational motion generator components consists of a single-rod mechanism, two-rod mechanism, a slider mechanism, a crankshaft mechanism, and others.

In previous research, a two-rod motion mechanism has been used to drive the generator components and produce an output voltage of 40 volts at 200 rpm rotation, as shown in Figure 1 [31–33]. This voltage can be increased by increasing the rotation of the generator; in previous studies, the thrust of ocean waves was able to push the generator translation up and down by 1400 rpm using a single-rod mechanism [29]. This shows that the output of the generator (voltage) can be increased by selecting a stiffer, smoother, and easier motion mechanism. A common problem in older power plants was that generator components were heavy, easy to break, less rigid, and had low rotation speed. Therefore, to overcome this problem, a generator with a two-rod mechanism is used in this research.

In addition to the selection of the motion mechanism, the performance of the translational and rotational motion generator is also influenced by the type, shape, and material of the magnet, the number and dimensions of the coil windings, the distance between the magnet and the coil windings, the type, dimensions, and shape of the generator motion mechanism, and other factors [31–33].

This study is focused on the shape of the combination of motion mechanisms between translational and rotational generators. The combination of this generator motion requires several mechanical components, namely shafts, bearings, bevel gears, and a two-rod mechanism. This motion mechanism drives translational and rotational motion generators simultaneously. The problem in this study was the motion mechanism, as it requires a large torque at the start of the movement of the mechanical components. This can be overcome by selecting or using the appropriate bevel gear for the generator's motion components. In this study, a bevel gear ratio of 1:5 was used to facilitate the movement of the translational and rotational motion generators, and the two-rod motion mechanism in the form of a round plate with a diameter of 200 mm and a square piston with a length of 270 mm is used as the driving medium for the translation and translation generators. Shafts and bearings are used as a seat and as a successor of rotation from the two-rod motion mechanism to the generator of translational and rotational motion. In this test, the

translational motion generator rotation was 100 rpm to 300 rpm. From the test results, the generator performance of translational and rotational motion was obtained in the form of voltage and electric power.

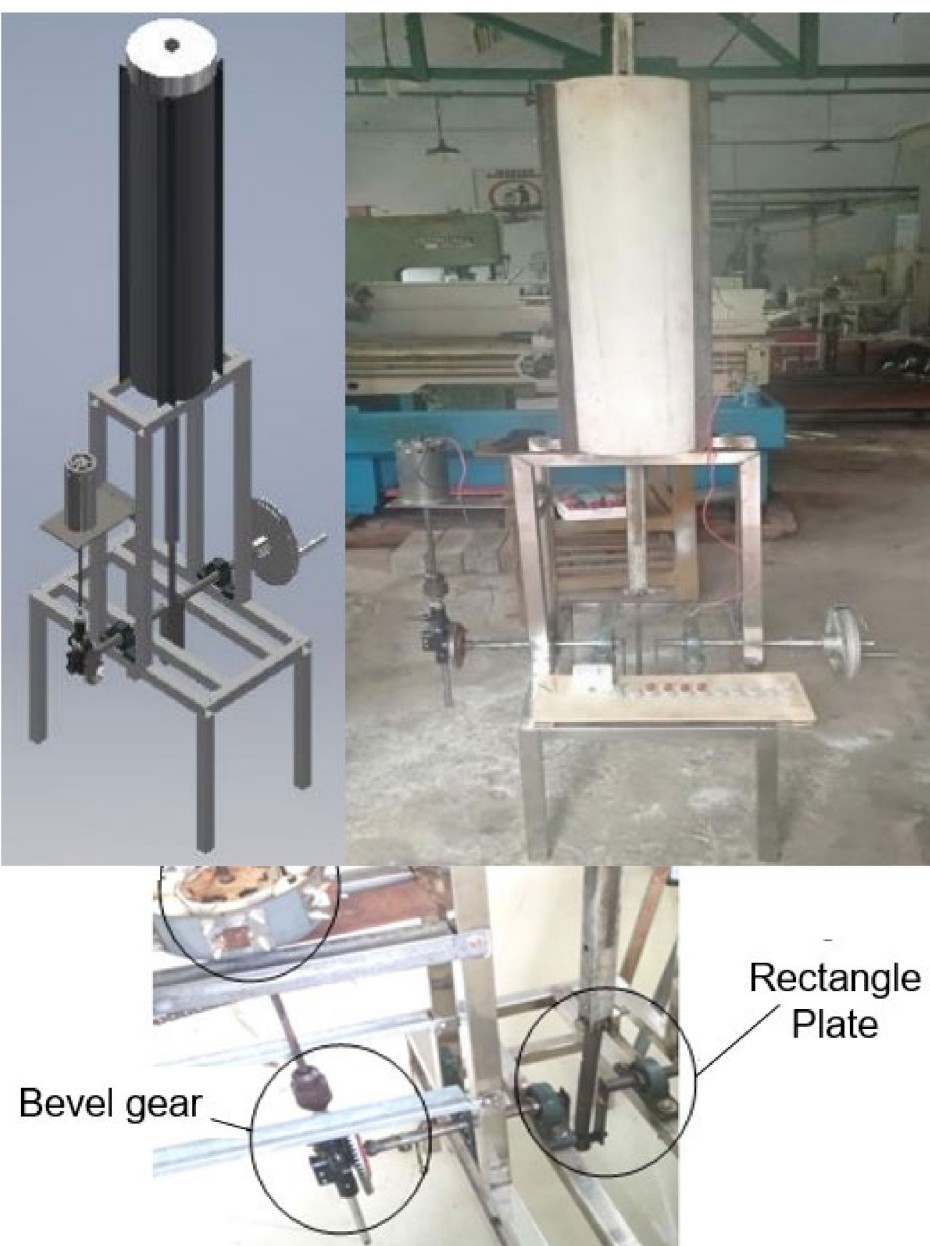

**Figure 1.** Previous study of generator translation and rotational motion.

The problem occurs when combining translational and rotational motion generators, which require a transmission mechanism to be able to work in a coupled manner. The two-rod mechanism has been used with a rotation of 200 rpm, but the generator movement is still heavy due to the condition of the two piston rods, which are less rigid, the low surface roughness values of components such as gears, and the smooth motion of shaft. In this study, we apply the crankshaft motion mechanism to facilitate the motion of the generator and also to increase the rigidity of the motion mechanism. The motion mechanism of the translational generator using the crankshaft will be coupled using a bevel gear to drive the rotational motion generator to obtain the output voltage of the translational and rotational motion generator.

## 2. Materials and Methods

### 2.1. Materials and Equipment

The translational and rotational motion generator using the two-rod motion mechanism consists of several components, namely translational and rotational generators, generator holder frames, shafts, bevel gears, bearings, two-rod motion mechanisms, and other things. The generator consists of a generator housing, a rotor containing a trapezium-shaped neodymium magnet, and a stator containing a coil winding. The number of magnets in the translational generator is 8, while there are 6 for the rotational generator. The number of turns of a copper wire coil with dimensions of 0.25 mm is 1260 turns for one coil. The number of coils is 4 for the translational generator and 6 for the rotational generator.

For the translation generator, the magnets are mounted on a square hollow rod wall of stainless steel material arranged with 4 on the left side and 4 on the right side. The coil winding is mounted on the wall of the generator housing, which is made of polyvinyl chloride (PVC) material. The generator housing is 254 mm in diameter and 80 mm in height. The generator housing is mounted on a generator holder frame made of stainless steel material. The square hollow rod sections are connected by pistons and round plates using bolts. The steel plates are connected by a shaft to the bevel gear with a ratio of 1:5 to unite with the rotational generator.

The rotating generator consists of 6 magnets mounted on a shaft of resin-coated wood. The magnet will contact the coil winding, which is attached to the wall of the polyvinyl chloride (PVC) material tube, which is 254 mm in diameter and 80 mm in height. The number of coil turns is 6, with the same number of turns as the coil turns on the translation generator, which is 1260 mm. The shape of the translational and rotational motion generator components can be seen in Figure 2. The series of translational and rotational motion generators can be seen in Figure 3.

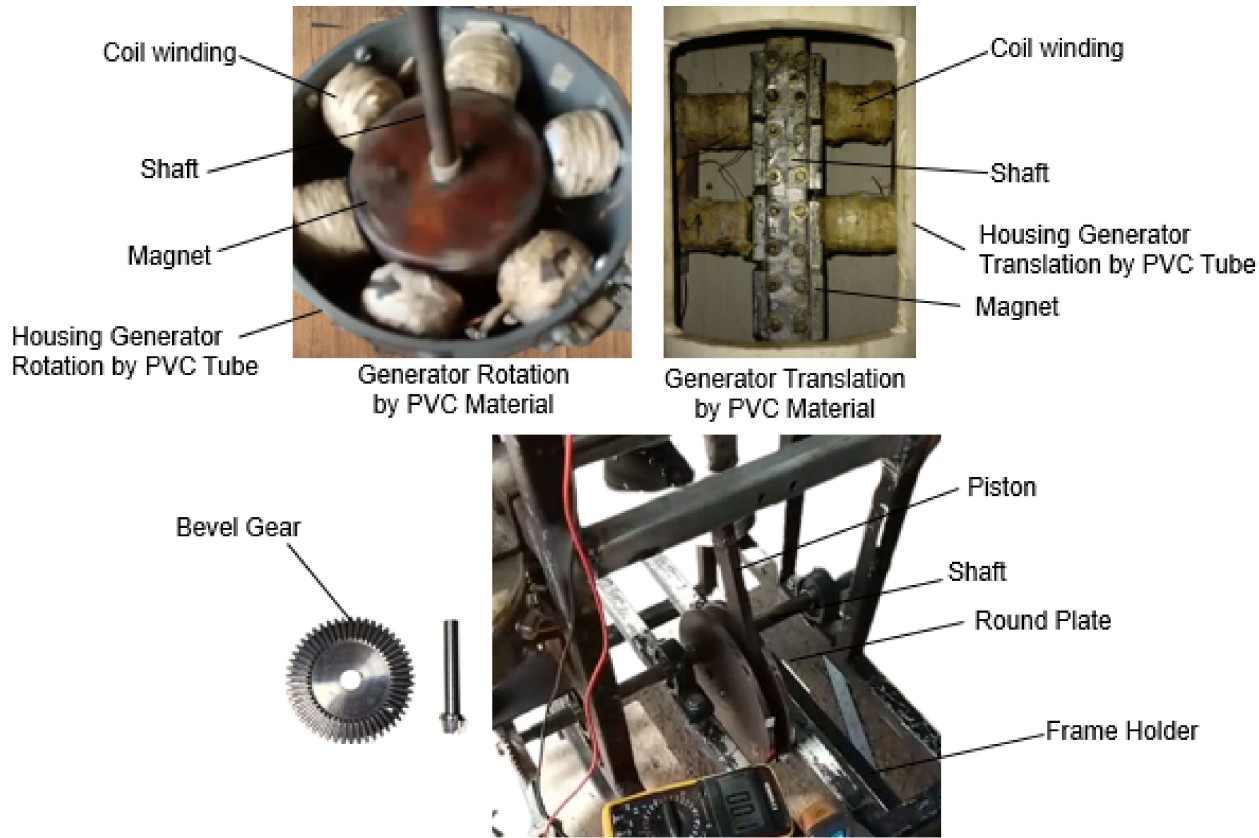

**Figure 2.** Component of generator translation and rotational motion.

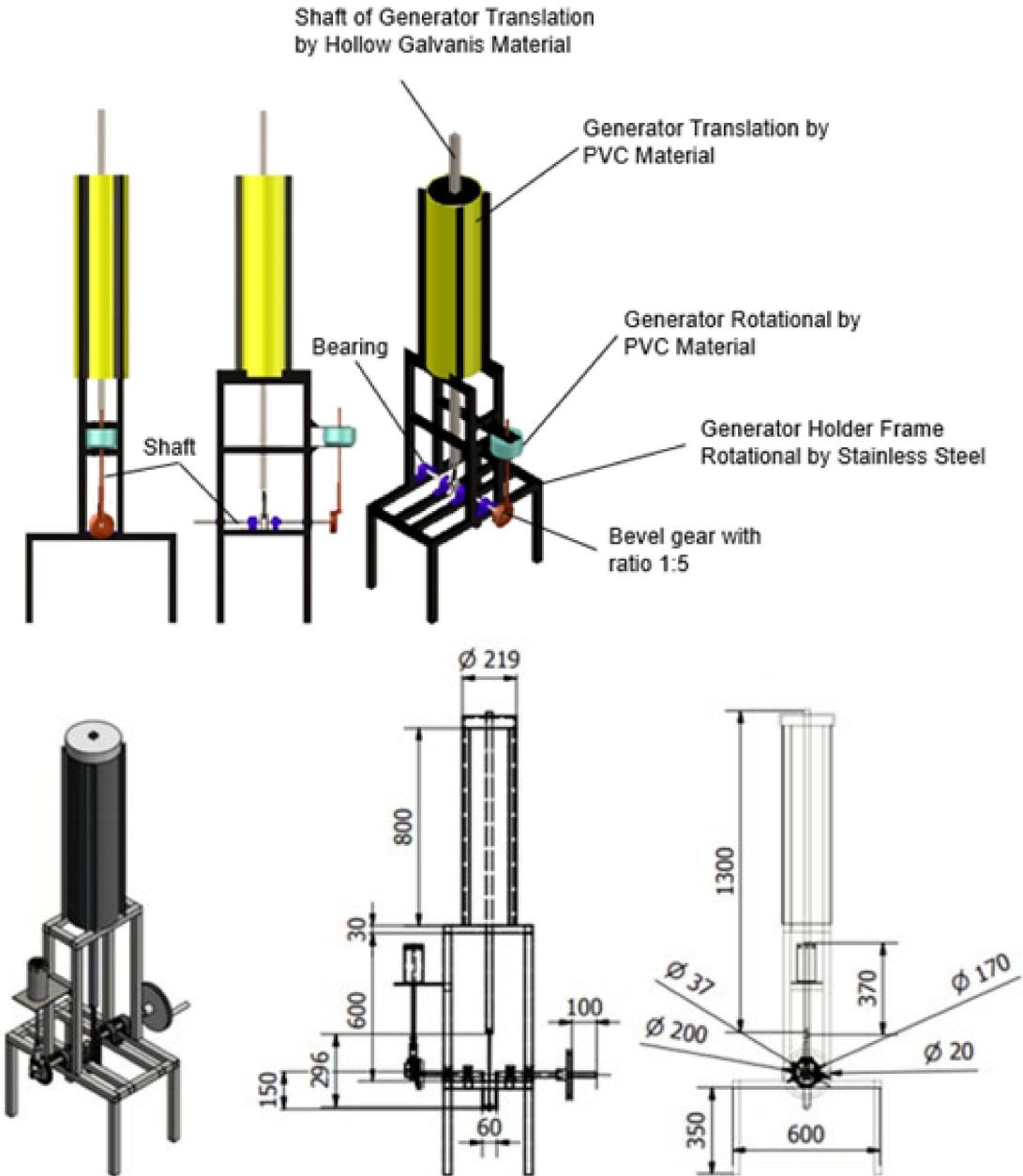

**Figure 3.** Construction of generator translation and rotational motion.

*2.2. Experimental Method*

This research was conducted by using an experimental method to obtain the performance of translational and rotational motion generators using a two-rod motion mechanism. The performance of translational and rotational motion generators is obtained in the form of rotation and output voltage. The output voltage of the generator is obtained from the rotation settings and the step distance of the crankshaft mechanism. The rotation speed used is 100 to 300 rpm for generator translation, and a gear ratio of 1:5 is used for generator rotation. The frequency of the generator can be obtained by using Equation (1) [34,35]. The step distance of the two-rod motion mechanism is 170 mm, 230 mm, and 270 mm. By varying the rotation of the generator and the distance between the motions of the two rods, the output voltage of the generator is obtained [34,35].

$$N_s = \frac{120.f}{p} \; ; \; (\text{rpm}) \quad f = \frac{N_s.p}{120} \; ; \; (\text{Hz}) \tag{1}$$

$N_s$ = rotation of generator (rpm);
*f* = frequency of generator (Hz);
*p* = number of magnetic poles.

The stages of the research included testing the translational and rotational generators separately with variations in speed, testing the combined translational and rotational generators, and testing the translational and rotational generators in series and parallel connections.

### 2.3. Set Up and Testing of Generator Translation and Rotational Motion

In carrying out the tests, we first pre-prepared the translational and rotational motion generator test equipment using a two-rod mechanism. This included the preparation of test equipment, including checking translational and rotational motion generators and measuring instruments, such as a tachometer to measure rotation and a voltmeter to measure voltage. Then, we measured the speed and output voltage of translational, rotational, and combined translational and rotational generators in motion. We connected the translational and rotational motion generators in series and parallel, measured the generator voltage, and calculated the power of the generator using Equation (2) [36].

$$P = V.I; \text{ (watt)} \tag{2}$$

*P* = power of generator (rpm);
*V* = voltage of generator (V);
*I* = electric current (A).

## 3. Results

Rotation and electric voltage are the results of testing the performance of translational and rotational motion generators using a two-rod motion mechanism, which are tested separately, loaded, and in series and parallel connections. The results of the translational and rotational motion generator tests obtained the output in the form of rotation from 78 to 955 rpm, and the electric voltage generated by the generator was 0.02–79.5 volts, as shown in Tables 1–6.

### 3.1. Results of Performance Generator Translational and Rotational Motions with Stroke Length 170 mm, 230 mm, and 270 mm

The results of translational and rotational motion generator performance tests using a two-rod motion mechanism with stroke lengths of 170 mm, 230 mm, and 270 mm, which were tested separately between translational and rotational motion generators, can be seen in Tables 1–3. Table 1 shows the amount of rotation produced by the electric voltage generator of translational and rotational motion with a piston length of 170 mm. The results of testing generators separately show that the rotation obtained is 102 to 191 rpm for translational motion generators and 510–955 rpm for rotational motion generators. The rotation of the generator by means of a motion test using a two-rod mechanism can be carried out up to 400 rpm for translational motion generators and 1700 rpm for rotational motion generators. In this paper, for piston lengths of 170 to 270 mm, data are only given up to 200 rpm for translational motion generators and 1000 rpm for rotational motion generators. The electric voltage obtained is 30.9 to 35.8 volts at 6.8–12.7 Hz for translational motion, while for rotational motion generators, these values are 0.5–0.9 V and 34–63.7 Hz. The magnitude of the value of the output voltage of the generator for translational motion has increased with increasing rotation, namely 30.9 V at 102 rpm rotation and 35.9 V at 191 rpm rotation. In contrast to the rotational motion generator, the increase in generator rotation is not followed by an increase in the output voltage of the generator but only produces the same voltage value between 0.5 and 0.9 V at 510–955 rpm, as shown in Table 1.

**Table 1.** Results of performance of generator translational and rotational motions with stroke length 170 mm.

| Performance of Generator | | | | | |
|---|---|---|---|---|---|
| Rotation of Generator Translational Motion (Rpm) | Rotation of Generator Rotational Motion (Rpm) | Voltage of Generator Translational Motion (V) | Frequency of Generator Translational Motion (Hz) | Voltage of Generator Rotational Motion (V) | Frequency of Generator Rotational Motion (Hz) |
| 102 | 510 | 30.9 | 6.8 | 0.5 | 34.0 |
| 118 | 600 | 31.9 | 7.9 | 0.3 | 40.0 |
| 180 | 900 | 34.8 | 12.0 | 0.9 | 60.0 |
| 191 | 955 | 35.8 | 12.7 | 0.5 | 63.7 |

After testing the performance of translational and rotational motion generators using a two-rod motion mechanism with a stroke length of 230 mm, the results obtained are a rotation of 78–172 rpm for translational motion generators and 390–860 rpm for rotational motion generators, as shown in Table 2. The generated electric voltage by a translational motion generator is 46.7–55.5 volts at 5.2–11.5 Hz, and the output voltage of this generator has a higher value compared to using a two-rod motion mechanism with a stroke length of 170 mm (see Tables 1 and 2). For the performance of the rotational motion generator, it has an electric voltage value of 0.4–0.5 volts at 19.5–43 Hz at 390–860 rpm.

**Table 2.** Results of performance of generator translational and rotational motions with stroke length 230 mm.

| Performance of Generator | | | | | |
|---|---|---|---|---|---|
| Rotation of Generator Translational Motion (Rpm) | Rotation of Generator Rotational Motion (Rpm) | Voltage of Generator Translational Motion (V) | Frequency of Generator Translational Motion (Hz) | Voltage of Generator Rotational Motion (V) | Frequency of Generator Translational Motion (Hz) |
| 78 | 390 | 46.7 | 5.2 | 0.5 | 19.5 |
| 82 | 407 | 47.6 | 5.5 | 0.4 | 20.3 |
| 168 | 840 | 48.9 | 11.2 | 0.5 | 42.0 |
| 172 | 860 | 55.5 | 11.5 | 0.5 | 43.0 |

In testing the translational motion generator using a two-rod mechanism with a stroke length of 270 mm, the generator performance value was obtained in the form of an electric voltage of 39.9–55.5 volts at 11.5–17.6 Hz at 172–256.5 rpm, as shown in Table 3. Meanwhile, for the electric voltage on the rotational motion generator with the same mechanism, the value of the electric voltage was obtained from 0.03 volts to 0.5 volts at 37–46 Hz at 739–922 rpm rotation.

**Table 3.** Results of performance of generator translational and rotational motion with stroke length 270 mm.

| Performance of Generator | | | | | |
|---|---|---|---|---|---|
| Rotation of Generator Translational Motion (Rpm) | Rotation of Generator Rotational Motion (Rpm) | Voltage of Generator Translational Motion (V) | Frequency of Generator Translational Motion (Hz) | Voltage of Generator Rotational Motion (V) | Frequency of Generator Translational Motion (Hz) |
| 172 | 860 | 55.5 | 11.5 | 0.5 | 43.0 |
| 209.2 | 922.2 | 39.9 | 13.9 | 0.03 | 46.1 |
| 241.9 | 739.9 | 41.8 | 16.1 | 0.04 | 37.0 |
| 256.5 | 739.9 | 40.7 | 17.1 | 0.02 | 37.0 |

This shows that for a translational generator, an increase in rotation and stroke length will be followed by an increase in the generator voltage up to 55.5 volts. Meanwhile, the output voltage from the rotational motion generator has almost the same value and even decreases at a piston length of 270 mm.

### 3.2. Results of Performance Generator Translation and Rotation with Load LED Lamp

We tested the performance of translational and rotational motion generators using a two-rod motion mechanism with a light-emitting diode (LED) lamp load and a stroke length of 270 mm; the generator rotation results were 200.1–235.6 rpm, as shown in Table 4. The electric voltage generated by the translational motion generator and the rotation is 29–37.7 volts, and the electric current is 0.4–1.3 A. From the results of this load test, it was observed that the electric power of the generator is 15.1–37.7 w. The value of this electric power depends on the load used, the voltage generated, and the current obtained.

From the results of testing the generator for translational and rotational motion using the light-emitting diode (LED) lamp load, it can be seen that the higher the rotation, the greater the voltage generated. However, the current generated by the generator is inversely proportional to the voltage, which is lower, as shown in Table 4. In Table 4, it can be seen that at 200.1 rpm for the translation generator and 1000 rpm for the rotational generator, we obtained an electric voltage of 29 Volts, an electric current of 1.3 mA, and 0.377 watts. When the rotation increases to 235.6 rpm, the electric voltage generated by the generator is 37.7 volts, the electric current is 0.4 mA, and the electric power is 0.151 watts. This shows that there is a decrease in electric power caused by a decrease in electric current, even though the rotation has increased.

**Table 4.** Results of performance of generator translational and rotational motion with load light-emitting diode (LED) Lamp.

| Performance of Generator | | | | |
|---|---|---|---|---|
| Rotation of Generator Translational Motion (Rpm) | Rotation of Generator Rotational Motion (Rpm) | Voltage of Generator Translational and Rotational Motion (V) | Current of Generator Translational and Rotational Motion (mA) | Power of Generator Translational and Rotational Motion (w) |
| 200.1 | 1000 | 29 | 1.3 | 0.377 |
| 217.6 | 1088 | 35.5 | 1 | 0.355 |
| 220 | 1100 | 36.9 | 0.9 | 0.332 |
| 235.6 | 1178 | 37.7 | 0.4 | 0.151 |

### 3.3. Results of Performance Generator Translation and Rotation in Series

Testing on the performance of generators of translational and rotational motion using a two-rod motion mechanism in series with a stroke length of 270 mm obtained a rotation from 179.2 rpm to 242.3 rpm and an electric voltage from 57.4 volts to 79.5 volts. From the results of testing the series connection generator, it can be seen that the output voltage is directly proportional to the increase in rotational and translational motion generator rotation, as shown in Table 5.

**Table 5.** Results of performance of generator translational and rotational motion in series.

| Performance of Generator | |
|---|---|
| Rotation of Generator Translational and Rotational Motion (Rpm) | Voltage of Generator Translational and Rotational Motion (V) |
| 179.2 | 57.4 |
| 202.5 | 62 |
| 203.3 | 63.3 |
| 205.3 | 65.3 |
| 209 | 71.9 |
| 241 | 77.5 |
| 242.3 | 79.5 |

*3.4. Results of Performance Generator Translation and Rotation in Parallel*

Testing the performance of translational and rotational motion generators using a two-rod motion mechanism in parallel with a stroke length of 270 mm obtained a rotation from 194.2 rpm to 246.2 rpm and an electric voltage from 36.1 volts to 44.5 volts, as shown in Table 6. From the test results of this parallel connection generator, it can be seen that the output voltage has almost the same value at the mains voltage using a translational motion generator.

**Table 6.** Results of performance of generator translational and rotational motion in parallel.

| Performance of Generator | |
|---|---|
| **Rotation of Generator Translational and Rotational Motion (Rpm)** | **Voltage of Generator Translational and Rotational Motion (V)** |
| 194.7 | 36.6 |
| 207.5 | 36.4 |
| 213.2 | 36.1 |
| 222.9 | 43.9 |
| 246.2 | 44.7 |

## 4. Discussion

The results of testing the performance of generators of translational and rotational motion using a two-rod mechanism with a gear ratio of 1:5 show that the mechanism of motion of this generator can work simultaneously. The resulting rotation is in accordance with the 1:5 ratio selection plan, namely from 78 rpm for the translational generator and 390 rpm for the rotational motion generator. The rotation produced by the generator seems to be able to increase the generated electric voltage, especially in translational motion generators.

In testing the length of the piston for the two-rod motion mechanism on the translational and rotational motion generator, an increase was shown in the rotation and voltage of the generator, especially in the translational motion generator. The test results show that the piston length has an effect on the performance of the translational motion generator, where the longer the piston, the higher the resulting rotation. High rotation can produce a large electric voltage, as shown in Tables 1–3. This large electric voltage is obtained because the contact position between the magnet and the coil winding is almost perfect, reaching the peak of the position of the magnet and the coil winding, as shown in Figure 4. The problem with increasing the length of the piston is that the torque required is large for the initial propulsion of the generator.

In a rotational motion generator, the resulting increase in rotation is not followed by an increase in the electric voltage. The value of the electric voltage generated by the rotational motion generator tends to have the same value of 0.5 volts (constant). This is due to the large gap between the magnet and the coil winding, which is 5 mm (see Figure 5). In addition to the gap between the magnet and the coil, the condition of the rotating shaft, which is less rigid and stable, results in a small and variable voltage value, especially in a rotational generator using a two-rod motion mechanism with a piston length of 270 mm where the generated voltage ranges from 0.02 volts to 0.5 volts. This problem can be overcome by adjusting the distance between the magnetic gap and the coil winding and using a rigid and strong shaft design. From setting the distance between the magnetic gap and the coil winding at 3 mm, the electric voltage for the rotational generator reaches 17.1 volts at 249.7 rpm, as shown in Table 7. This result shows that with a small gap between the magnet and the coil, the performances of generator rotational motion can be increased, as shown in Figure 6.

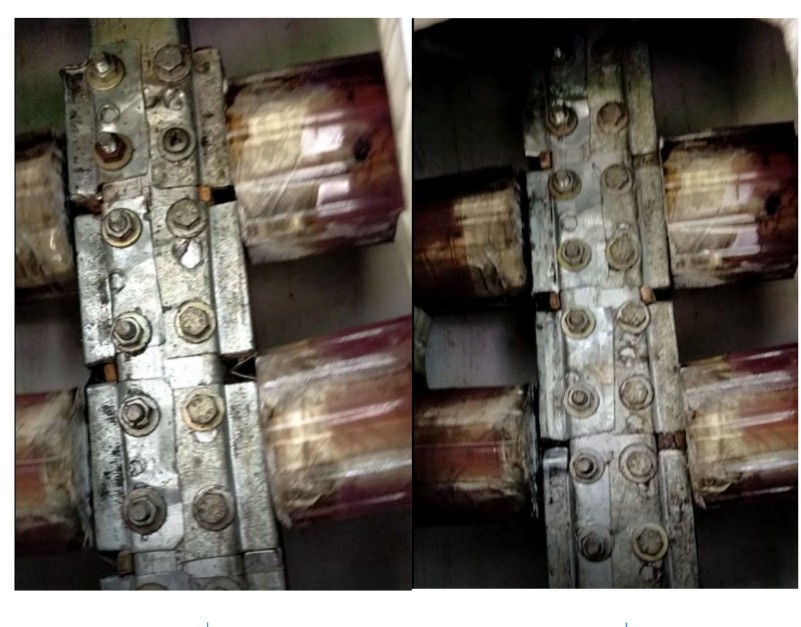

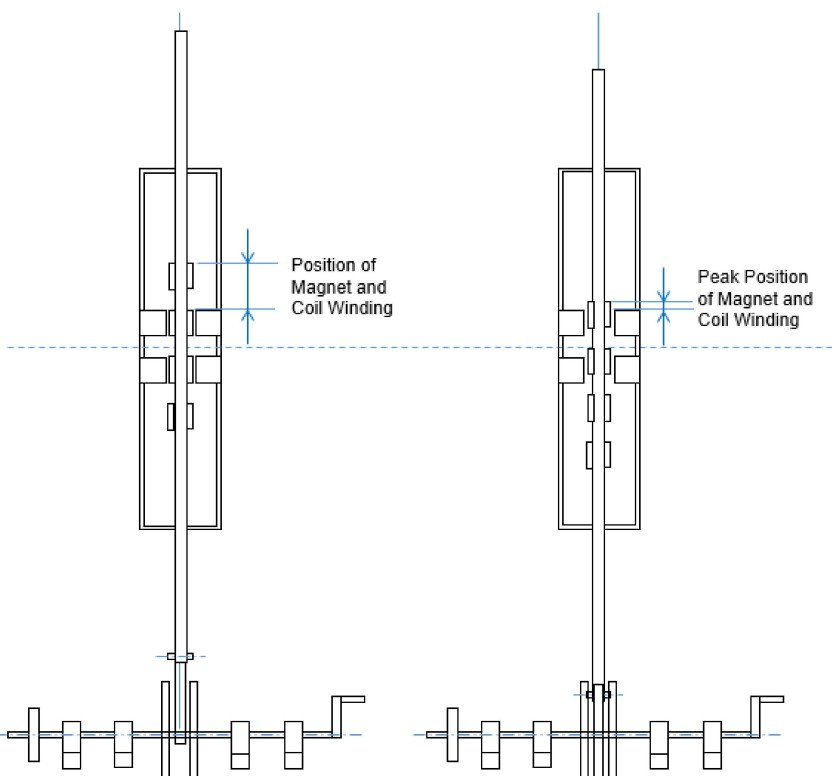

**Figure 4.** Position of magnet and winding coil of generator translation and rotational motion.

**Table 7.** Result of performance of generator translational and rotational motion with rotation 138–256 rpm.

| Performance of Generator | | | |
|---|---|---|---|
| Rotation of Generator Translational Motion (Rpm) | Rotation of Generator Rotational Motion (Rpm) | Voltage of Generator Translational Motion (V) | Voltage of Generator Rotational Motion (V) |
| 122.8 | 547.7 | 22.4 | 14.2 |
| 208 | 1188 | 22.2 | 12.5 |
| 240.6 | 1188 | 22.5 | 17.1 |
| 250 | 1188 | 23.9 | 17.1 |

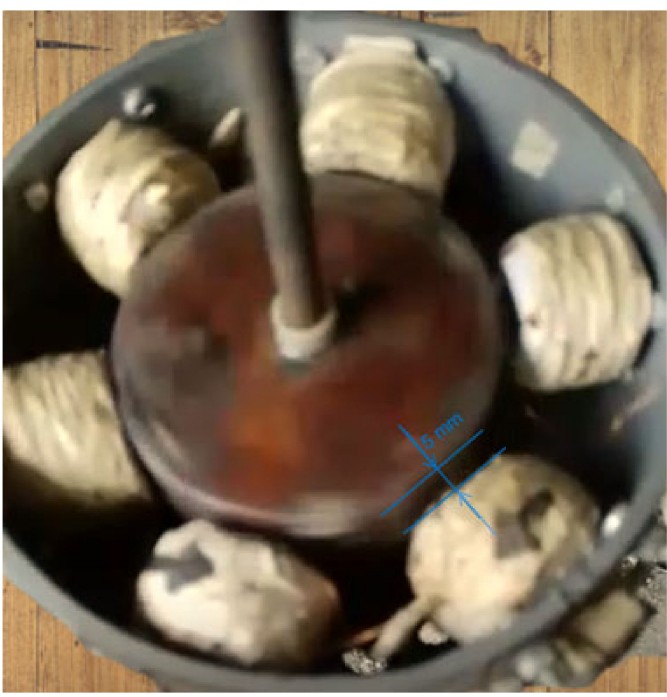

**Figure 5.** 5 mm gap of magnet and winding coil of generator translation and rotational motion.

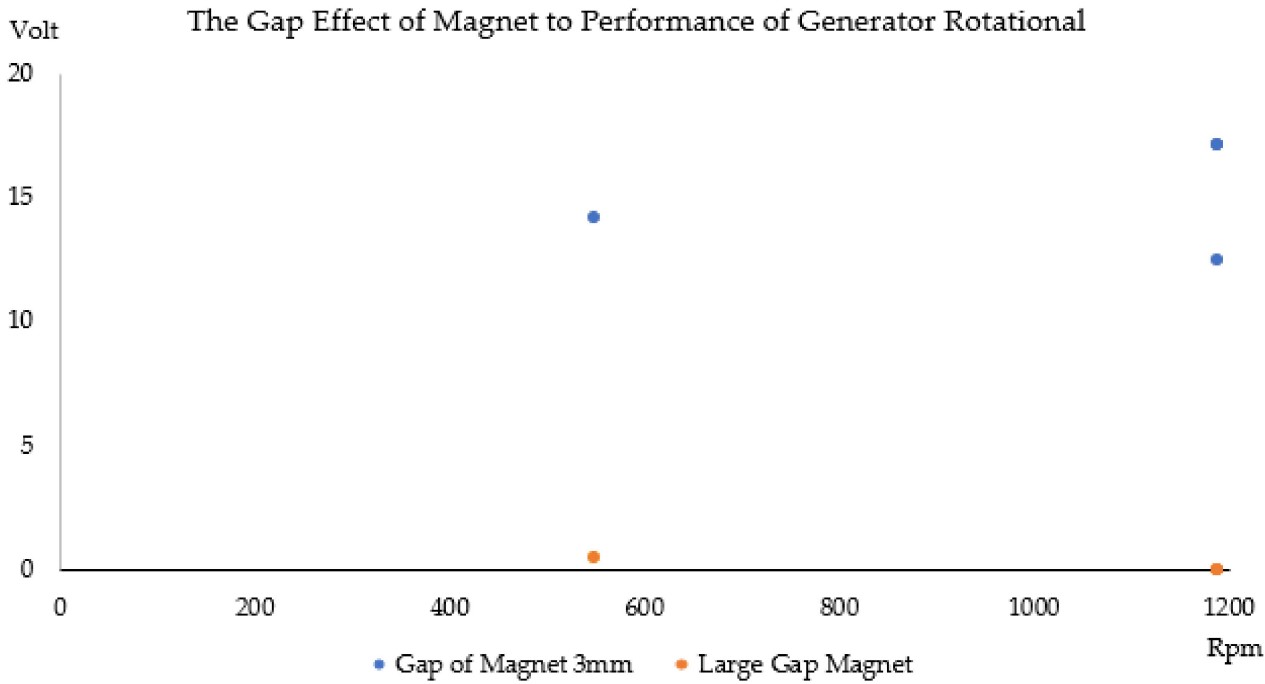

**Figure 6.** The effect of gap between magnet and winding coil on generator rotational motion.

### 5. Conclusions

Testing of the performance of translational and rotational motion generators using a two-rod mechanism for the length of the piston, it can be concluded that the generator with two-rod motion mechanism at a rotation of 100–250 rpm produces an electric voltage of 30.9–55 volts at a frequency of 6.9–63.7 Hz, with a maximum power of 0.377 watts. By setting a piston stroke length of 170 mm, we obtained a rotation of 100–191 rpm and an electrical voltage of 30.9–35 volts. At a piston stroke length of 230 rpm, a rotation of 78–172 rpm is obtained with an electrical voltage of 47.7–55.5 volts. A piston stroke length

of 270 mm obtains a rotation of 172–256.5 rpm with a mains voltage of 39.9–55.5 volts. Testing the generators of translational and rotational motion using a two-rod motion mechanism in series and in parallel with a stroke length of 270 mm obtained a rotation from 179.2 rpm to 242.3 rpm and an electric voltage from 57.4 volts to 79.5 volts, and this voltage reached a constant 35.6 volts using the parallel mechanism. The two-rod-driven mechanism can increase the rotation and electric voltage generated by the generator. The greater the length of the piston, the greater the rotation and the generated electric voltage, namely 256 rpm and 55 volts in the translational motion generator. Unlike the translational motion generator, the testing results of the rotational motion generator showed a constant voltage value of 0.5 volts at 500–1000 rpm rotation. This is due to the gap distance between the magnet and the rotational generator coil winding, which is rather large at 5 mm. By using a small magnetic gap and a coil winding of 3 mm, the electric voltage of the rotational motion generator reached 17.1 volts at a rotation of 1188 rpm. In the load testing, the generator produced 0.377 watts of electric power, with a current of 1.3 mA, and 29 volts. From the load test it can be seen that the greater the voltage and the smaller the current, the smaller the electric power produced. For tests of translational and rotational motion generators with serial and parallel connections, it can be seen that the electric voltage increases with an increase in rotation of the series-connected generators, but for generators with parallel connections, the resulting electric voltage is almost the same as the magnitude of the electric voltage on the translational motion generator. The results in this research show that the generator translation and rotation motion can produce electric power by using renewable energy resources with a two-rod mechanism.

**Author Contributions:** Conceptualization, H.H. (Hendra Hendra), H.H. (Hernadewita Hernadewita); methodology, H.H. (Hendra Hendra), D.S., H.H. (Hernadewita Hernadewita) and Y.Y.; validation data, H.H. (Hendra Hendra), D.S., H.H. (Hernadewita Hernadewita) and Y.Y.; formal analysis, H.H. (Hendra Hendra), H.H. (Hernadewita Hernadewita) and Y.Y.; investigation, H.H. (Hendra Hendra), H.H. (Hernadewita Hernadewita) and Y.Y.; resources, H.H. (Hendra Hendra), D.S., H.H. (Hernadewita Hernadewita), Y.Y. and F.H.; data curation, H.H. (Hendra Hendra).; writing—original draft preparation, H.H. (Hendra Hendra), H.H. (Hernadewita Hernadewita) and F.H.; writing—review and editing, H.H. (Hendra Hendra), H.H. (Hernadewita Hernadewita), F.H. and A.M.G.; supervision, H.H. (Hendra Hendra), H.H. (Hernadewita Hernadewita), and A.M.G. All authors have read and agreed to the published version of the manuscript.

**Funding:** This research was financially supported by the Ministry of Education, Culture, Research and Technology Republic of Indonesia through Hibah Penelitian Dasar Unggulan Perguruan Tinggi Number B/439/43.9/PT.00.03/2022 and Institution of Research and Community Service University of Sultan Ageng Tirtayasa Banten Indonesia. Also this paper is supported via funding from Prince Sattam Bin Abdulaziz University Project Number (PSAU/2023/R/1444).

**Institutional Review Board Statement:** Not applicable.

**Data Availability Statement:** Not applicable.

**Conflicts of Interest:** The authors declare no conflict of interest.

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
