# Peer review of "Performance of Generator Translation and Rotation on Stroke Length Drive of the Two-Rod Mechanism in Renewable Energy Power Plant"

_sustainability, doi:10.3390/su15075663_

Round 1

Reviewer 1 Report

This paper is very interesting and relevant. Renewable energy is very widely used these days. Mankind needs new types of generators to convert the energy of sea waves and tides. Therefore, the study of the authors is very interesting.

But there are some comments on the presentation of materials. The article does not contain mathematical formulas. There are no charts in the article.

I think that it is necessary to strengthen the mathematical component of the article. You need to add math formulas. And build graphs of changes in the parameters of the generator.

Author Response

Answer for reviewer question.

  1. Math formula have been add as suggestion in Page 5 Line 8 and Line 24 from top sentence.
  2. Revision done see figure 6.

Reviewer 2 Report

Reviewer Comments

In this paper, generators are the main components in renewable energy power plants, especially in 17 ocean wave of power plants energy. Generator consists of two components of translational and ro-18 tational motion. Generators of translational and rotational motion can produce power electric from 19 renewable energy sources such as water, wind, sea waves, biomass and others. The voltage and 20 electric power are the performance values of the translational and rotational generators which are 21 affected by the type of magnet, the number of coil winding, the distance between the magnet and 22 the coil winding and rotation, the geometry of the drive components, the type of drive, the length 23 of the generator drive stroke and so on. The followings should be carefully addressed in the revision to be published in Sustainability.

1-      The authors should be followed the instruction of the Sustainabilityall parts and sections in this manuscript.

2-      Complete mathematic calculation model with all nomenclature missing

3-      The abstract needs more quantitative results. The abstract section is an important and powerful representation of the research. It is better that the results should be presented with the support of specified data. Please provide your contribution and work novelty.

4-      The authors should indicate this technique to enhance system performance. Also, the author should add more references that discuss the effect of using this technique. It is recommended that the authors carry out wide analysis and comparison with the state-of-the-art studies.

5-      Most tables and figures are needed improve the quality of all tables and figures.

6-      Add references for all equations.

7-      I would also expect to validate with two more experimental works available in the literature.

8-      The literature review must be improved. Please highlight in the literature review the differences between previous papers and your paper. Please clearly indicate the knowledge gap and prove that it is a really not analyzed area of the field. Please indicate new approach / new methods in a comparison to the existing investigations (literature review should be extended by adding the below references). The Effect of Different Twisted Tape Inserts Configurations on Fluid Flow Characteristics, Pressure Drop, Thermo-hydraulic Performance and Heat Transfer Enhancement in the 3D Circular Tube.

9-      You need to add error analysis of your results and add the error bars in your graphs to indicate your accuracy measurements.

10-  Improve work justification.

11-  More quantitative conclusions should be presented. Please prepare additional comparisons, some percentage differences. There is a lack of quantitative conclusions which should contain main findings from the paper and highlight the new and high novelty and contribution of your work to the field.

12-  Present the mathematical equation of the boundary conditions and initial condition.

13-  I would also suggest including in the conclusion section but also in several other places in the manuscript discussion and comparison with findings from other authors with similar published research work.

14-  The conclusion section on lacks in summative conclusions. The main results, novelty and academic contributions should be emphasized in this section. Moreover, are the results obtained in this paper really applicable in other similar researches?

15-  In the discussion development, it is very important to emphasize points of agreement or disagreement between results in this work and others cited in references part of manuscript.

16-  Authors should discuss limitations of the current study and possible improvements for future directions/research works.

17-  The nomenclature list is not complete. Please recheck parameters, variables and abbreviations appeared in the manuscript and append them to the nomenclature list.

18-  Finally, the author to read through the whole text and correct it to make it more reader-friendly.

Author Response

Answer for reviewer question.

  1. The authors have followed all the instruction of MDPI-Sustainability.
  2. Math formula have been add as suggestion in Page 5 Line 8 and Line 24 from top sentence.
  3. Abstract have been put some more qualitative results as request in abstract page 1 line 12 from top abstract.
  4. This research by DOE (design of experiment) technique applied to this research. Past of DOE the experiments were made by produced and testing of machine generator. This experiment was 2nd Stage and this generator is new model.
  5. Revision done
  6. Revision done
  7. Rotation, voltage, current and power of generator have been validation by testing and references.
  8. Revision done.
  9. Revision done figure 6.
  10. Revision done.
  11. Revision done in Page 11 line 4 from top sentence.
  12. This research by experimental and by using equation 1 and 2 to calculated the frequency and power of generator.
  13. Revision done
  14. Revision done
  15. Revision done
  16. Revision done
  17. Revision done
  18. Revision done

Round 2

Reviewer 2 Report

Reviewer Comments

The followings should be carefully addressed in the revision to be published in Sustainability.

1-      The abstract still needs more quantitative results. The abstract section is an important and powerful representation of the research. Please provide your contribution and work novelty.

2-      Most tables and figures are still needed improve the quality of all tables and figures.

3-      Add references for all equations.

4-      The literature review must be improved. Please highlight in the literature review the differences between previous papers and your paper. Please clearly indicate the knowledge gap and prove that it is a really not analyzed area of the field. Please indicate new approach / new methods in a comparison to the existing investigations (literature review should be extended by adding the below references). The Effect of Different Twisted Tape Inserts Configurations on Fluid Flow Characteristics, Pressure Drop, Thermo-hydraulic Performance and Heat Transfer Enhancement in the 3D Circular Tube.

5-      You need to add error analysis of your results and add the error bars in your graphs to indicate your accuracy measurements.

6-      Improve work justification.

7-      The conclusion section on lacks in summative conclusions.

8-      The nomenclature list is not complete.

Author Response

Answer for reviewer question.

  1. Abstract have been put some more qualitative results as request in abstract page 1 line 9 from bottom. (see line 9 from bottom).
  2. Tables and Figure have been more big add as suggestion in Page 3-5. For Table we used template from MDPI instruction.
  3. References have been put in page 4 line 1 from bottom and page 6 line 2 from top.
  4. The literature have been put at paragraph 4 line 4 from bottom paragraph.
  5. Due the experiment in this research, its applied min requirement test for the machine, as of 100-200 rpm (Anizar Indriani et al). For error test, we will conduct for the next research as the comparative method in machines testing and it will include validation test. It will applied in the 2nd year of our research.
  6. Revision done at page 4 line 6 from bottom.
  7. Revision done page 12 line 3 from top.

Revision done. The terminology had use in this paper suit with the basic concepts.
